# Motion-Aware Caching for Efficient Autoregressive Video Generation

**Jing Xu** [1 2 *]   **Yuexiao Ma** [1 *]   **Xuzhe Zheng** [1 *]   **Xing Wang** [2 †]   **Shiwei Liu** [3 4 5]   **Chenqian Yan** [2]   **Xiawu Zheng** [1]
**Rongrong Ji** [1]   **Fei Chao** [1]   **Songwei Liu** [2]

## Abstract

Autoregressive video generation paradigms offer theoretical promise for long video synthesis, yet their practical deployment is hindered by the computational burden of sequential iterative denoising. While cache reuse strategies can accelerate generation by skipping redundant denoising steps, existing methods rely on coarse-grained chunk-level skipping that fails to capture fine-grained pixel dynamics. This oversight is critical: pixels with high motion require more denoising steps to prevent error accumulation, while static pixels tolerate aggressive skipping. We formalize this insight theoretically by linking cache errors to residual instability, and propose **MotionCache**, a motion-aware cache framework that exploits inter-frame differences as a lightweight proxy for pixel-level motion characteristics. Motion-Cache employs a coarse-to-fine strategy: an initial warm-up phase establishes semantic coherence, followed by motion-weighted cache reuse that dynamically adjusts update frequencies per token. Extensive experiments on state-of-the-art models like SkyReels-V2 and MAGI-1 demonstrate that MotionCache achieves significant speedups of **6.28**$\times$ and **1.64**$\times$ respectively, while effectively preserving generation quality (VBench: $1\% \downarrow$ and $0.01\% \downarrow$ respectively). The code is available at https://github.com/ywlq/MotionCache.

## 1. Introduction

Video generation models (Ma et al., 2024; Yang et al., 2024; Zheng et al., 2024; Kong et al., 2024; Peng et al., 2025; Wan et al., 2025; Gao et al., 2025b) have achieved remarkable success, facilitating applications ranging from autonomous driving (Fu et al., 2024; Wen et al., 2024; Gao et al., 2025a) and cinematic creation (Xing et al., 2025; Chen et al., 2025) to other domains. While architectures have evolved from U-Nets (Saharia et al., 2022; Ramesh et al., 2022; Blattmann et al., 2023) to scalable Diffusion Transformers (DiTs) (Peebles & Xie, 2023), practical deployment is hindered by the prohibitive costs of iterative denoising. Moreover, the quadratic complexity of attention mechanisms regarding video resolution and duration imposes strict memory limits, creating severe bottlenecks for real-time adoption.

To address these scalability limitations, autoregressive video generation models (Teng et al., 2025; Chen et al., 2025) leverage the Causal Diffusion-Forcing (CDF) framework (Chen et al., 2024a; Yin et al., 2025; Song et al., 2025; Yang et al., 2025) to adapt the next-token prediction paradigm to the video domain. Unlike full-sequence methods, these models decouple memory usage from total video duration by partitioning the stream into chunks and utilizing KV cache. This strategy effectively reduces attention complexity from quadratic to linear, keeping memory consumption bounded and theoretically permitting infinite generation. Nevertheless, despite these structural optimizations, the autoregressive generation of high-resolution and long-duration videos remains inherently time-consuming. For example, using an $A800$ GPU with batch size 1, generating a 7-second video at $540 \times 540$ resolution requires approximately 27 minutes of inference time for the SkyReels-V2 model.

To mitigate the computational burden of iterative denoising, caching-based strategies have emerged as pivotal solutions. By exploiting the high temporal redundancy inherent in the diffusion process to reuse intermediate features or residuals, these methods are broadly categorized into layer-level and step-level approaches. Layer-level methods (Selvaraju et al., 2024; Zou et al., 2024; Zhao et al., 2024b; Cui et al., 2025) cache intermediate feature maps but necessitate storing features for every layer. This results in memory consumption that grows linearly with model depth, an overhead further

---

[*]Equal contribution [†]Project leader [1]Key Laboratory of Multimedia Trusted Perception and Efficient Computing, Ministry of Education of China, Xiamen University, 361005, P.R. China [2]ByteDance [3]Max Planck Institute for Intelligent Systems [4]ELLIS Institute Tbingen [5]Tbingen AI Center. Correspondence to: Fei Chao <fchao@xmu.edu.cn>, Songwei Liu <liusongwei.zju@bytedance.com>.

*Proceedings of the $43^{rd}$ International Conference on Machine Learning*, Seoul, South Korea. PMLR 306, 2026. Copyright 2026 by the author(s).

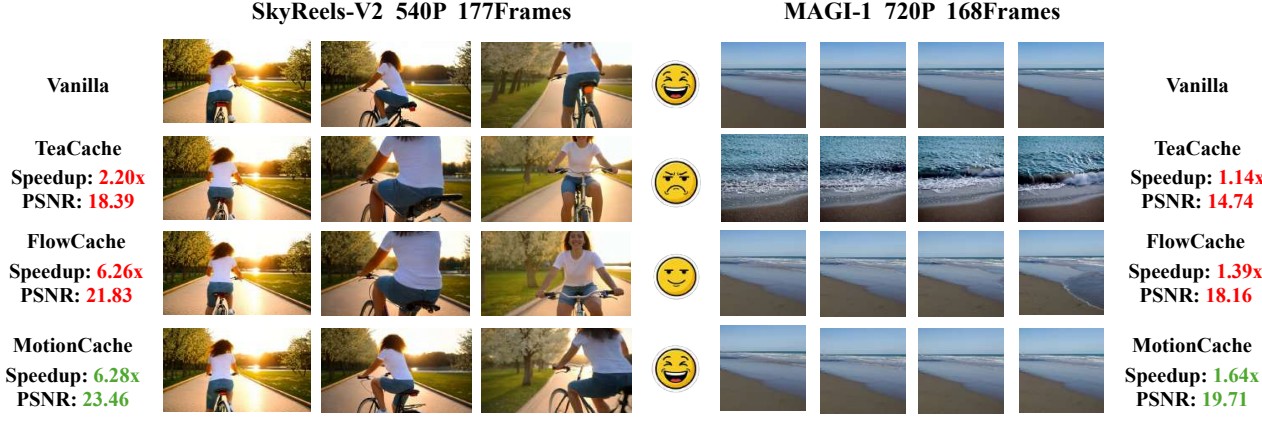

*Figure 1.* MotionCache accelerates video generation while maintaining high visual fidelity. On SkyReels-V2 and MAGI-1, our method achieves 6.28× and 1.64× speedups with superior PSNR. In contrast, TeaCache fails to maintain texture details and FlowCache suffers from structural inconsistency, while MotionCache preserves both structural integrity and temporal coherence comparable to the Vanilla baseline.

exacerbated by advanced predictors using Taylor expansions (Liu et al., 2025b; Zheng et al., 2025; Liu et al., 2025c). Conversely, step-level methods (Liu et al., 2025a; Kahatapitiya et al., 2025; Yu et al., 2025; Bu et al., 2025) store residuals for reuse in skipped steps. While offering negligible memory overhead, their prediction precision often falls short compared to finer-grained layer-level techniques.

Crucially, these existing approaches are primarily tailored for standard DiTs and do not generalize to autoregressive video generation models. FlowCache (Ma et al., 2026) represents the first attempt to bridge this gap by exploiting the temporal heterogeneity of autoregressive chunks. However, FlowCache relies on a coarse-grained binary strategy where an entire chunk must simultaneously be computed or skipped. This approach overlooks fine-grained token-level temporal redundancy and fails to account for the significant spatial and content discrepancies between different frames within the same timestep.

To address these limitations, we present MotionCache, a motion-aware caching framework grounded in a theoretical analysis linking caching error to residual instability. Leveraging the intra-chunk frame difference as a lightweight, high-fidelity proxy for pixel-level motion characteristics, MotionCache operationalizes a hierarchical coarse-to-fine inference schedule. This mechanism initially secures global structural integrity through a warm-up phase, and subsequently transitions to a token-wise adaptive policy that dynamically allocates computational resources—prioritizing updates for high-motion regions while efficiently retrieving cached residuals for static backgrounds. Through extensive experimentation, we demonstrate that our approach not only achieves superior acceleration ratios but also preserves the

quality of generated videos.

In summary, our key contributions are as follows:

- We provide a rigorous theoretical analysis of the approximation error in feature caching, identifying that the error is strictly bounded by residual instability. We further uncover the fundamental Heterogeneous Temporal Redundancy and Intra-Chunk Frame Discrepancy in autoregressive models, demonstrating the diverse update requirements across different frames and tokens that traditional coarse-grained strategies overlook.

- We establish a theoretical link between residual stability and video motion dynamics, proving that the frame difference serves as a mathematically grounded upper bound for caching error. Based on this, we introduce motion-aware token importance, a lightweight and high-fidelity proxy that enables precise identification of dynamic regions.

- We propose MotionCache, a novel motion-aware acceleration framework that implements a coarse-to-fine inference schedule. By synergizing a structural warm-up phase with a motion-characteristics-weighted token accumulation policy, MotionCache dynamically allocates computational resources, prioritizing high-motion details.

- Extensive evaluations on state-of-the-art autoregressive video generation models, including SkyReels-V2 and MAGI-1, demonstrate that MotionCache significantly outperforms existing methods. It achieves speedups of 7.26× and 2.07× respectively, while delivering superior visual fidelity and temporal coherence. These results establish MotionCache as the new state-of-the-art for efficient autoregressive video generation.

## 2. Related Work

### 2.1. AutoRegressive Video Generation

Our work builds upon recent advances in autoregressive video generation models (Teng et al., 2025; Chen et al., 2025), which fundamentally integrate the next-token prediction paradigm of Large Language Models (LLMs) into the video synthesis process. In this framework, the continuous video stream is discretized into a sequence of video chunks, which are generated sequentially based on Causal Diffusion-Forcing (CDF) (Chen et al., 2024a; Yin et al., 2025; Song et al., 2025; Yang et al., 2025) inference paradigms. The underlying generator for each chunk typically employs a diffusion backbone optimized with flow matching objectives (Lipman et al., 2022). By leveraging a fixed attention window alongside this chunking technique, these methods effectively circumvent the quadratic computational complexity inherent in full-sequence modeling. This architecture not only ensures scalable efficiency but also achieves remarkable success in synthesizing high-fidelity video content with strong causal consistency and temporal coherence.

**Feature Caching-based Acceleration.** Feature caching accelerates inference by exploiting temporal redundancy in a training-free manner. Early adaptations like FORA (Selvaraju et al., 2024) and $\Delta$-DiT (Chen et al., 2024b) employed fixed reuse schedules, while recent advancements focus on dynamic, content-aware policies. Methods such as TeaCache (Liu et al., 2025a) and AdaCache (Kahatapitiya et al., 2025) estimate caching intervals based on input differences or video complexity, whereas TaylorSeer (Liu et al., 2025b) predicts feature trajectories via Taylor expansions. In the autoregressive domain, FlowCache (Ma et al., 2026) extends these concepts to chunk-level skipping. However, these methods predominantly operate at a coarse granularity—treating entire timesteps or chunks as atomic units—thereby failing to exploit fine-grained token-level redundancy and struggling to adapt to spatially heterogeneous motion dynamics.

## 3. Preliminaries

**Diffusion Model.** Diffusion models (Ronneberger et al., 2015; Peebles & Xie, 2023) synthesize data by reversing a noise injection process. Under the Flow Matching paradigm (Lipman et al., 2022), the forward transition from data $\pi_0$ to Gaussian prior $\pi_1$ follows a linear interpolation path:

$$\mathbf{x}_t = (1 - \sigma(t)) \cdot \mathbf{x}_{data} + \sigma(t) \cdot \mathbf{x}_{noise}, \quad (1)$$

where $\sigma(t)$ is a monotonic scheduling function. The reverse denoising phase recovers data by approximating a time-dependent velocity field $v_\theta$, governed by the ODE $d\mathbf{x}/dt = v_\theta(\mathbf{x}_t, t, c)$, where $c$ denotes conditional inputs. Inference

is typically performed using numerical solvers like Euler's method (Karras et al., 2022) to iteratively update the sample:

$$\mathbf{x}_{t_{i-1}} = \mathbf{x}_{t_i} + v_\theta(\mathbf{x}_{t_i}, t_i, c)\Delta t_i. \quad (2)$$

**AutoRegressive Video Generation Model.** To circumvent the quadratic complexity inherent in full-sequence modeling, the framework adopts an autoregressive generation paradigm by decomposing long videos into discrete units. Formally, a video sequence is partitioned into $k$ latent chunks, denoted as $\{\mathbf{X}^1, \dots, \mathbf{X}^k\}$. Each chunk $\mathbf{X}^i \in \mathbb{R}^{F \times H \times W \times C}$ represents a high-dimensional latent representation characterized by a temporal duration of $F$, a spatial resolution of $H \times W$, and $C$ channels. The generation of the $i$-th chunk is conditioned on the preceding context, where the denoising update at timestep $t$ is governed by the Euler (Karras et al., 2022) discretization step:

$$\mathbf{X}_{t-1}^i = \mathbf{X}_t^i + v_\theta(\mathbf{X}_t^i, t, c) \cdot \Delta t. \quad (3)$$

Here, $t \in [(i-1)T/l, (i+l-1)T/l]$ represents the timestep, where $T$ denotes the total number of discretization steps and $l$ indicates the maximum window size permitted for the denoising process.

**Feature Caching Strategies.** Feature caching accelerates inference by exploiting temporal redundancy. Caching decisions rely on metrics like the relative L1 distance (Liu et al., 2025a; Bu et al., 2025; Cui et al., 2025) between consecutive inputs $\mathbf{X}_t^i$:

$$L1_{rel}(\mathbf{X}^i, t) = \frac{\|\mathbf{X}_t^i - \mathbf{X}_{t+1}^i\|_1}{\|\mathbf{X}_{t+1}^i\|_1}. \quad (4)$$

During full computation, the system caches the residual, defined as the difference between the predicted velocity and input latent:

$$\mathcal{R}_t^i = v_\theta(\mathbf{X}_t^i, t, c) - \mathbf{X}_t^i. \quad (5)$$

When the accumulated relative L1 distance falls below the threshold, the forward pass is bypassed, and the velocity is approximated by reusing the cached residual:

$$\tilde{v}_{t-1}^i \approx \mathbf{X}_{t-1}^i + \mathcal{R}_t^i. \quad (6)$$

This mechanism effectively reduces computational overhead by leveraging local feature stability.

## 4. Analysis of Caching Error

To motivate our transition from coarse-grained chunk skipping to fine-grained token-wise caching, we analytically investigate the source of approximation error in the caching mechanism. We demonstrate that the caching error is theoretically bounded by the inconsistency of feature residuals across timesteps, and this inconsistency is highly correlated with the underlying motion dynamics.

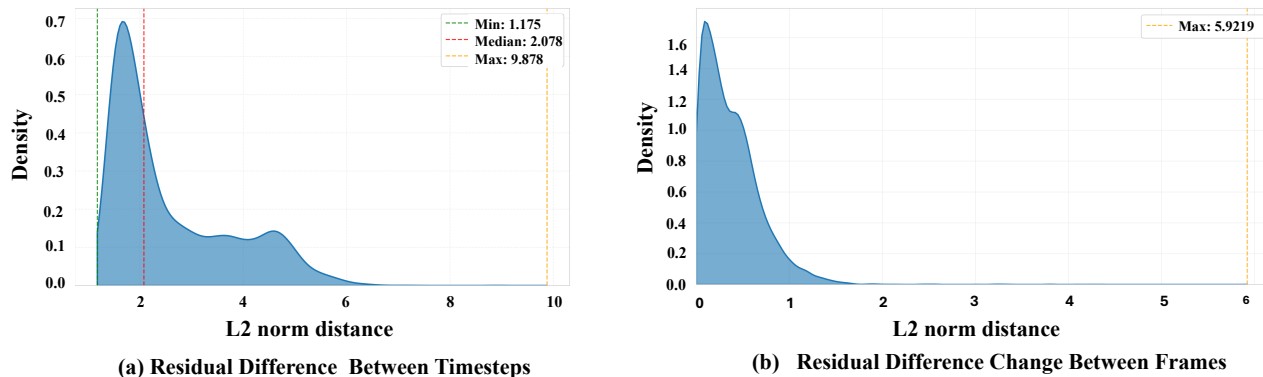

*Figure 2.* (a) **Heterogeneous Temporal Redundancy:** The distribution of residual differences between adjacent timesteps exhibits a long-tailed pattern. While the majority of tokens cluster around low values, a significant tail extends to high values, indicating highly non-uniform update requirements across tokens. (b) **Intra-Chunk Frame Discrepancy:** The distribution of residual changes across distinct frames within the same chunk reveals significant variation. This wide dynamic range confirms that frames within a single autoregressive chunk possess distinct motion characteristics, rendering coarse-grained cache suboptimal.

## 4.1. Theoretical Error Bound of Feature Caching

Consider the denoising process for the $i$-th video chunk at timestep $t - 1$. We quantify the *Local Approximation Error* $\epsilon_{t-1}^i$ as the Euclidean distance between the ground-truth output derived from full computation and the approximated output derived from residual reuse. Based on the flow-matching update rule, we derive the following relationship regarding the caching error:

**Proposition 4.1** (Residual Inconsistency Principle). *The approximation error at timestep $t - 1$ for chunk $i$ is strictly proportional to the magnitude of the vector difference between the true residual $\mathcal{R}_{t-1}^i$ and the cached residual $\mathcal{R}_t^i$:*

$$\epsilon_{t-1}^i = \Delta t \cdot \|\mathcal{R}_{t-1}^i - \mathcal{R}_t^i\|_2. \tag{7}$$

The detailed derivation is provided in Appendix A. This proposition provides a fundamental insight: the reliability of the caching mechanism is governed by the temporal stability of the residual term. A larger deviation between the reused residual and the current true residual leads directly to a larger caching error. Therefore, the optimal caching policy must selectively calculate tokens where $\|\mathcal{R}_{t-1}^i - \mathcal{R}_t^i\|$ is significant, while retrieving tokens where this difference is negligible.

## 4.2. Empirical Observations

Guided by Proposition 4.1, we analyze the distribution of residual differences in actual video generation.

**Heterogeneous Temporal Redundancy.** Figure 2 (a) presents the Kernel Density Estimation (KDE) of residual differences between adjacent timesteps. The resulting long-tailed distribution—characterized by a low median (2.078) yet a significant tail reaching 9.878—reveals that feature update requirements are highly non-uniform. This hetero-

geneity invalidates uniform chunk-wise caching, which inevitably wastes computation on static regions or degrades dynamic ones, thereby motivating the need for an adaptive token-wise strategy.

**Intra-Chunk Frame Discrepancy.** Figure 2 (b) illustrates the residual heterogeneity across distinct frames within a single autoregressive chunk. Since video VAEs compress multiple consecutive raw frames into a single latent frame, distinct latent frames correspond to different temporal segments of the original video. Consequently, the residual distribution extends significantly (max difference 5.9219), reflecting the varied content evolution across these segments. This substantial intra-chunk discrepancy confirms that frames are not uniformly redundant; thus, treating the entire chunk as an atomic unit is suboptimal, necessitating a more fine-grained update mechanism.

## 4.3. Theoretical Connection: Residual Stability and Motion Dynamics

Since the ideal residual difference is computationally inaccessible prior to inference, we require a lightweight proxy. We establish a theoretical bound linking the temporal instability of residuals to the spatial-temporal variations of the input latent.

**Lemma 4.2** (Motion-Induced Residual Instability). *Let $\mathcal{R}(X, t)$ be the continuous residual function derived from the velocity field. Assuming the temporal gradient of the residual field $\nabla_t \mathcal{R}$ satisfies the Lipschitz condition with respect to the input latent $X$, the residual difference across timesteps is bounded by the intra-chunk frame difference:*

$$\|\mathcal{R}_{t-1}(\mathbf{X}_{t-1}^{(i,f)}) - \mathcal{R}_t(\mathbf{X}_t^{(i,f)})\|_2 \lesssim C \cdot \|\mathbf{X}_t^{(i,f)} - \mathbf{X}_t^{(i,f-1)}\|_2, \tag{8}$$

*where $\mathbf{X}_t^{(i,f)}$ and $\mathbf{X}_t^{(i,f-1)}$ denote the latents of the $f$-th*

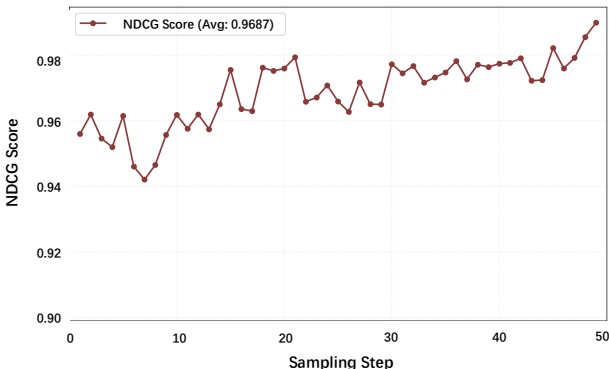

*Figure 3.* **Validation of Motion Proxy.** NDCG (Wang et al., 2013; Järvelin & Kekäläinen, 2017) scores comparing frame difference-based token importance rankings to rankings derived from adjacent timestep residual differences. The scores remain consistently above 0.94, demonstrating strong similarity in token importance ordering throughout the diffusion process.

*and $(f-1)$-th frames in the $i$-th chunk at timestep $t$, and $C$ is a constant.*

The detailed proof is provided in Appendix B. Equation 8 implies that the frame difference is not merely a heuristic, but a mathematically grounded upper bound for residual instability.

**Validation of the Motion Proxy.** To empirically validate this correlation, we treat the caching decision as a ranking problem. We compare the token importance ranking derived from our proposed frame difference against the ground-truth ranking derived from the actual residual difference. As illustrated in Figure 3, the Normalized Discounted Cumulative Gain (NDCG) (Wang et al., 2013; Järvelin & Kekäläinen, 2017) scores consistently exceed 0.94 across denoising timesteps. This demonstrates that the frame difference preserves the relative order of token importance with high fidelity, confirming it as an effective surrogate to precisely identify critical tokens for computation while retrieving stable ones from the cache.

## 5. Methodology

Based on the theoretical analysis in Sec. 4, we propose a fine-grained, motion-aware caching strategy tailored for autoregressive video generation , as illustrated in Figure 4. Our method dynamically allocates computational resources by prioritizing tokens in high-motion regions while efficiently reusing residuals for static backgrounds.

### 5.1. Motion-Aware Token Importance

The core of our strategy lies in accurately identifying tokens that require frequent updates. As derived in Equation 8, the intra-chunk frame difference serves as a robust proxy

for residual instability. Let $\mathbf{X}_t^i \in \mathbb{R}^{F \times H \times W \times C}$ denote the latent of the $i$-th video chunk at denoising timestep $t$, containing $F$ frames. We define the importance map $\mathcal{M} \in \mathbb{R}^{F \times H \times W}$ based on the token-wise difference between adjacent frames.

For a specific frame $f$ within chunk $i$, the importance score $\mathcal{M}_t^{(i,f)}$ is computed using the output latent from the previous timestep $t+1$:

$$\mathcal{M}_t^{(i,f)} = \begin{cases} \|\mathbf{X}_{t+1}^{(i,f)} - \mathbf{X}_{t+1}^{(i,f-1)}\|_1 & \text{if } f > 0, \\ \|\mathbf{X}_{t+1}^{(i,0)} - \mathbf{X}_{t+1}^{(i-1,F-1)}\|_1 & \text{if } f = 0 \text{ and } i > 0, \\ \mathcal{M}_t^{(0,1)} & \text{if } f = 0 \text{ and } i = 0. \end{cases}$$
(9)

Here, standard frames ($f > 0$) calculate the difference with their preceding frame. The first frame of a chunk ($f = 0, i > 0$) computes the difference with the last frame of the previously generated chunk ($i-1$) to maintain temporal continuity. For the very first frame of the entire video ($f = 0, i = 0$), which lacks a temporal reference, we reuse the importance score of the second frame.

To convert this raw importance into a modulation weight for caching, we apply a soft-mapping function based on min-max normalization. Crucially, this operation is performed independently for each frame to adapt to the varying dynamic ranges of motion across different temporal moments. We first normalize the importance scores $\mathcal{M}$ within the specific frame $f$ to the range $[0, 1]$, and then linearly project them to a target interval $[\alpha, 1]$:

$$\mathcal{W}_t^{(i,f)} = \alpha + (1-\alpha) \cdot \frac{\mathcal{M}_t^{(i,f)} - \min(\mathcal{M}_t^{(i,f)})}{\max(\mathcal{M}_t^{(i,f)}) - \min(\mathcal{M}_t^{(i,f)}) + \epsilon},$$
(10)

where $\min(\mathcal{M}_t^{(i,f)})$ and $\max(\mathcal{M}_t^{(i,f)})$ denote the spatial minimum and maximum importance values strictly within the current frame $f$, and $\epsilon$ is a small constant for numerical stability. The parameter $\alpha \in [0, 1]$ serves as a floor value, ensuring that even static background tokens (where $\mathcal{W} \approx \alpha$) continue to accumulate update probability at a baseline rate rather than being completely frozen.

Qualitatively, as illustrated in Figure 5, the computed importance maps demonstrate precise spatial correspondence with the ground-truth video frames, effectively distinguishing dynamic regions from static backgrounds.

### 5.2. Importance-Weighted Accumulation Policy

To dynamically determine the update frequency for each token, we introduce a weight-based accumulation mechanism. We track an error metric $\mathcal{A}$ for every spatial-temporal token location, which accumulates the estimated residual change over time.

At each timestep $t$, we first calculate the relative L1 distance

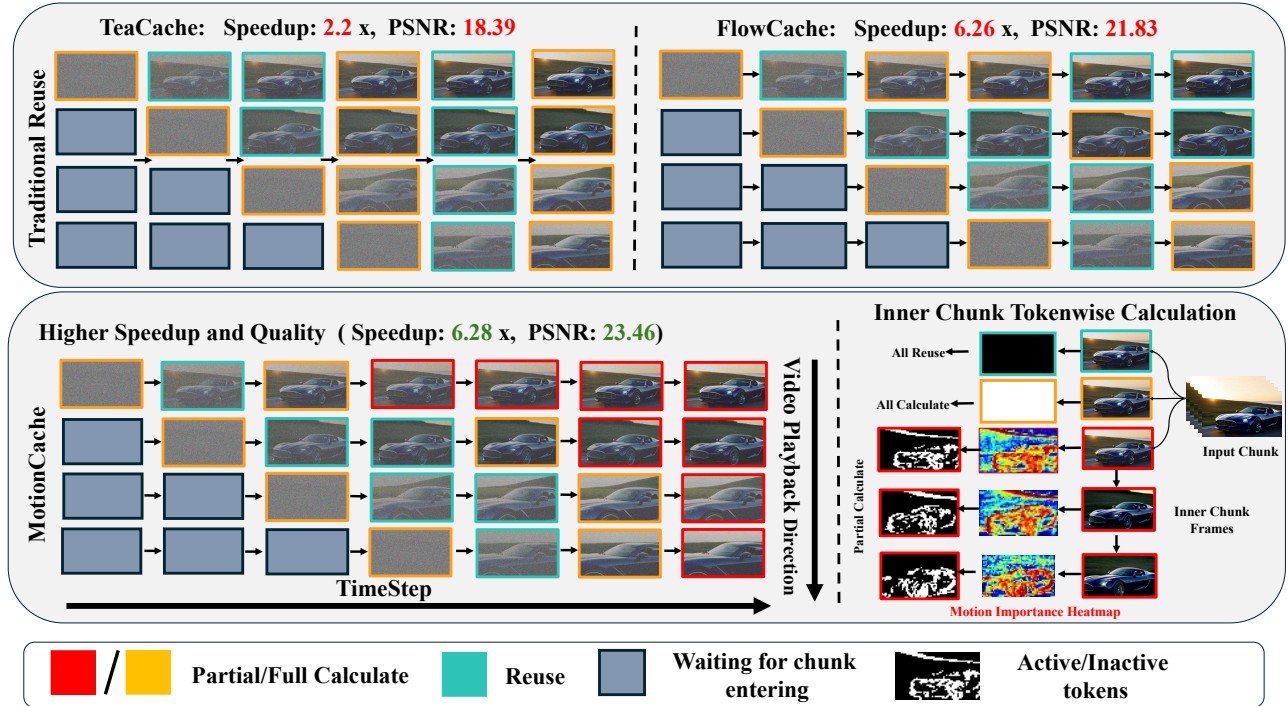

*Figure 4.* Comparison of caching strategies in autoregressive video generation. The top panel illustrates traditional reuse strategies (e.g., TeaCache and FlowCache), which apply coarse-grained caching policies by treating an entire timestep or chunk as an atomic unit for skipping. This approach overlooks fine-grained intra-chunk redundancy, forcing a binary decision between full computation or full reuse. In contrast, our MotionCache (bottom panel) employs a fine-grained Motion-Aware caching policy, dynamically deciding for each individual token whether to reuse cached residuals or perform recomputation based on motion dynamics. The bottom right panel details the Inner Chunk Tokenwise Calculation mechanism: it first calculates motion importance based on intra-chunk frame differences, then applies an importance-weighted accumulation policy to generate a binary selection mask, where white regions indicate active tokens selected for computation and black regions denote inactive tokens retrieved from the cache.

for the $i$-th chunk, denoted as $\Delta_{chunk}$, to represent the overall magnitude of the latent update:

$$\Delta_{chunk}(t) = \frac{\|\mathbf{X}_t^i - \mathbf{X}_{t+1}^i\|_1}{\|\mathbf{X}_{t+1}^i\|_1}. \quad (11)$$

Then, we distribute this update budget to individual tokens based on their motion-aware weights. The accumulator for a token at position $p$ is updated as:

$$\mathcal{A}_t[p] = \mathcal{A}_{t+1}[p] + \mathcal{W}_t[p] \cdot \Delta_{chunk}(t). \quad (12)$$

This strategy effectively couples the temporal denoising progress with spatial motion dynamics. High-motion tokens ($\mathcal{W} \approx 1$) absorb the full change and accumulate error rapidly, while static background tokens ($\mathcal{W} \approx \alpha$) suppress the accumulation. A token is selected for computation only when its accumulator exceeds a predefined threshold $\tau$:

$$\text{Mask}_t[p] = \mathbb{I}(\mathcal{A}_t[p] > \tau). \quad (13)$$

where $\mathbb{I}(\cdot)$ denotes the indicator function that takes the value 1 if the condition holds and 0 otherwise. Upon selection, the token undergoes a forward pass, and its accumulator $\mathcal{A}_t[p]$ is reset to 0.

## 5.3. Dual-Stage Coarse-to-Fine Inference Schedule

Video generation typically exhibits a coarse-to-fine progression, where global structures are established early, and high-frequency details are refined later (Teng et al., 2025; Lou et al., 2024; Kahatapitiya et al., 2025). To align with this property, we implement a dual-stage coarse-to-fine inference schedule.

**Phase 1: Coarse-grained Structure Construction.** In the initial phase of generation, maintaining global structural integrity is paramount. As illustrated in Figure 3, the NDCG scores exhibit significant volatility during these early timesteps, indicating that the semantic foundation is not yet stabilized.

Consequently, selective token forward at this stage could disrupt the formation of consistent semantic layouts. Therefore, we enforce a chunk-wise decision policy. At each step, the decision to compute or cache is synchronized across the entire chunk: the mask is effectively binary for the whole feature map (Mask $\in \{\mathbf{0}, \mathbf{1}\}$). The chunk is either fully updated or fully skipped. This phase continues until the model has performed a total of $K$ full computations, where $K$

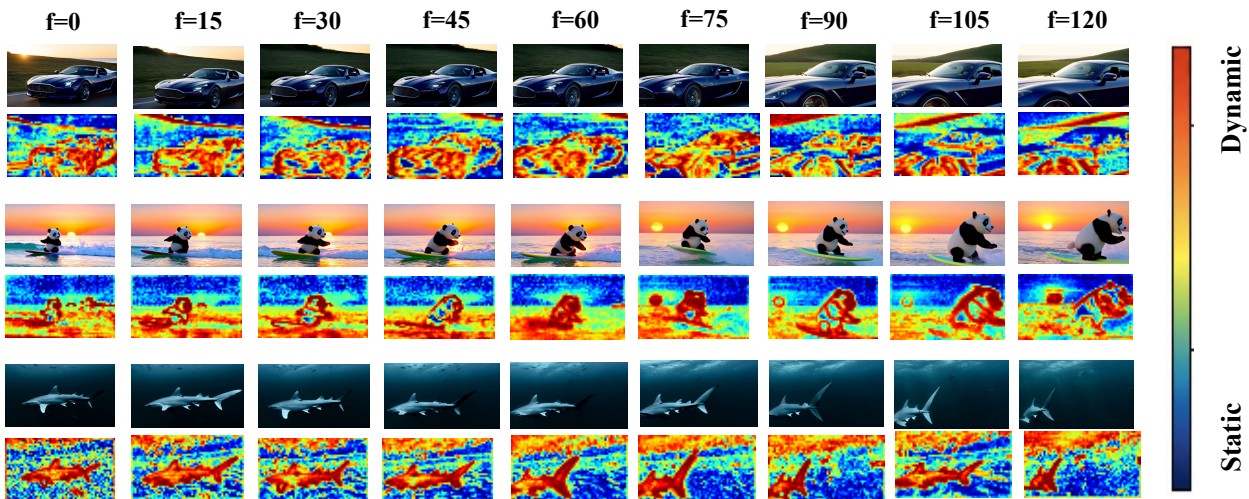

*Figure 5.* Visualization of the ground-truth video frames versus the computed importance maps. The label $f$ indicates the frame index within the video sequence.

is a hyperparameter controlling the structural foundation's solidity.

**Phase 2: Fine-grained Detail Refinement.** Once global structure stabilizes after $K$ full-chunk updates, we transition to the sparse token-wise adaptive mode described in Sec. 5.2. Leveraging the native KV cache, we gather only active tokens ($\text{Mask}[p] = 1$) into a compact batch for the forward pass. These computed features are subsequently scattered back to update the residual cache $\mathcal{R}_{cache}$, while inactive tokens bypass computation by directly retrieving stored residuals for approximation.

# 6. Experiment

## 6.1. Experimental Setup

**Base Models.** To evaluate the efficacy of our proposed method, we selected two representative diffusion models based on the autoregressive paradigm: MAGI-1-4.5B-distill (Teng et al., 2025) and SkyReels-V2-1.3B (Chen et al., 2025). For MAGI-1, we generate videos at 720p resolution consisting of 7 chunks, where each chunk contains 24 frames at 24 FPS. For SkyReels-V2, the generation targets a resolution of 540p, producing videos composed of 2 chunks with 97 frames each at 24 FPS.

**Evaluation Metrics.** Following established acceleration protocols such as FlowCache (Ma et al., 2026) and Tea-Cache (Liu et al., 2025a), we assess performance based on perceptual quality and computational efficiency. For quality, we employ standard metrics including LPIPS (Zhang et al., 2018), PSNR (Hore & Ziou, 2010), and SSIM (Wang et al., 2004). Furthermore, we utilize the VBench-long benchmark (Huang et al., 2024) for comprehensive video generation as-

sessment; for brevity, we refer to this as VBench throughout the paper. Efficiency is quantified by measuring Floating Point Operations (FLOPs) and practical inference latency.

**Implementation Details.** All experiments are implemented in PyTorch and executed on NVIDIA A800 80GB GPUs. Further details regarding the model implementation, along with a detailed introduction and specific configurations of the evaluation metrics, are provided in Appendix C.

## 6.2. Main Result

As shown in Table 1, MotionCache achieves superior efficiency-quality trade-offs compared to TeaCache and FlowCache. On MAGI-1, while TeaCache-fast and FlowCache-fast suffer significant quality degradation (VBench scores dropping to 68.81% and 73.42% respectively), MotionCache-fast achieves a 2.07× speedup while maintaining robust visual quality (VBench 74.59%). MotionCache-slow delivers nearly lossless quality with a 1.64× acceleration, effectively preserving fine-grained semantic details that are otherwise lost in coarse-grained schemes.

The advantage is even more pronounced on SkyReels-V2. MotionCache-slow achieves a 6.28× acceleration with a VBench score of 82.84%, significantly outperforming FlowCache-slow (6.26×, 82.70%) and TeaCache-slow (1.89×, 82.67%) in both speed and structural alignment (PSNR 23.46). MotionCache-fast maintains excellent quality (VBench 82.75%) at a state-of-the-art 7.26× speedup, whereas existing methods exhibit noticeable texture drifting and structural misalignment at significantly lower acceleration ratios(more visualization resluts in Appendix H).

*Table 1.* Quantitative comparison with state-of-the-art acceleration methods on SkyReels-V2 and MAGI-1. "Slow" and "Fast" denote configurations with lower and higher acceleration ratios, respectively. **MotionCache** achieves superior speedups while maintaining higher generation quality compared to other baselines.

| Model | Method | PFLOPs ↓ | Speedup ↑ | Latency(s) ↓ | VBench ↑ | PSNR ↑ | SSIM ↑ | LPIPS ↓ |
|---|---|---|---|---|---|---|---|---|
| SkyReels-V2 | Vanilla | 113 | 1× | 1540 | 83.84% | - | - | - |
| | TeaCache-slow | 58 | 1.89× | 814 | 82.67% | 21.96 | 0.7501 | 0.1472 |
| | TeaCache-fast | 49 | 2.2× | 686 | 80.06% | 18.39 | 0.6121 | 0.3063 |
| | FlowCache-slow | 31 | 6.26× | 246 | 82.70% | 21.83 | 0.8733 | 0.1417 |
| | FlowCache-fast | 27 | 7.19× | 214 | 82.38% | 21.17 | 0.8697 | 0.1634 |
| | MotionCache-slow | 30 | 6.28× | 245 | **82.84%** | **23.46** | **0.9093** | **0.0875** |
| | MotionCache-fast | **26** | **7.26×** | **212** | 82.75% | 21.78 | 0.8723 | 0.1478 |
| MAGI-1 | Vanilla | 139 | 1× | 1520 | 77.26% | - | - | - |
| | TeaCache-slow | 129 | 1.14× | 1339 | 76.64% | 14.74 | 0.4132 | 0.6189 |
| | TeaCache-fast | 101 | 1.41× | 1075 | 68.81% | 11.98 | 0.2632 | 0.7670 |
| | FlowCache-slow | 104 | 1.39× | 1094 | 77.08% | 18.16 | 0.6486 | 0.3451 |
| | FlowCache-fast | 78 | 1.94× | 782 | 73.42% | 14.92 | 0.3998 | 0.6088 |
| | MotionCache-slow | 100 | 1.64× | 925 | **77.25%** | **19.71** | **0.7231** | **0.2510** |
| | MotionCache-fast | **64** | **2.07×** | **733** | 74.59% | 17.70 | 0.5600 | 0.4861 |

## 6.3. Ablation Study

To investigate the efficacy of our design choices, we analyze the impact of two pivotal hyperparameters: the soft-mapping floor $\alpha$ and the Phase 1 duration $K$. The complete ablation tables detailing the numerical results for all experimental settings are provided in Appendix D.

**Impact of Soft-mapping Parameter $\alpha$.** As shown in Table 2, $\alpha = 0$ disables forced updates for static areas, while $\alpha = 1$ eliminates spatial selectivity, degenerating the method to FlowCache. Increasing $\alpha$ enhances background preservation by raising the update frequency for static tokens, though at the cost of higher latency. Empirically, $\alpha = 0.6$ strikes the optimal balance between quality and efficiency.

**Impact of Phase 1 Duration $K$.** As shown in Table 3, increasing $K$ extends the chunk-wise policy, eventually degenerating to FlowCache. While larger $K$ benefits global structure, it raises latency. Results indicate $K = 6$ is optimal; further increasing $K$ yields marginal quality gains while adding computational overhead.

## 7. Conclusion

In this paper, we presented MotionCache, a novel motion-aware caching framework designed to accelerate autoregressive video generation. By establishing a theoretical connection between residual instability and intra-chunk frame discrepancies, we introduced a lightweight, fine-grained proxy for token importance. This formulation allows the model to break free from the rigid "all-or-nothing" constraints of pre-

*Table 2.* Ablation study on the soft-mapping floor parameter $\alpha$.

| $\alpha$ | PSNR ↑ | SSIM ↑ | LPIPS ↓ |
|---|---|---|---|
| 0.0 | 20.22 | 0.7944 | 0.5853 |
| 0.2 | 21.82 | 0.8591 | 0.3663 |
| 0.4 | 22.90 | 0.8950 | 0.1744 |
| 0.6 | **23.46** | 0.9093 | 0.0875 |
| 0.8 | 23.42 | 0.9095 | **0.0862** |
| 1.0 | 23.44 | **0.9105** | 0.0866 |

*Table 3.* Ablation study on the duration of Phase 1 ($K$).

| $K$ | PSNR ↑ | SSIM ↑ | LPIPS ↓ |
|---|---|---|---|
| 0 | 20.79 | 0.8636 | 0.1963 |
| 3 | 22.92 | 0.8993 | 0.1027 |
| 6 | **23.46** | 0.9093 | 0.0875 |
| 9 | 23.39 | 0.9090 | 0.0883 |
| 12 | 23.41 | 0.9092 | **0.0859** |
| 15 | 23.43 | **0.9103** | 0.0865 |

vious coarse-grained methods, enabling dynamic resource allocation that prioritizes high-motion regions while efficiently reusing residuals for static backgrounds. Extensive experiments on state-of-the-art models, SkyReels-V2 and MAGI-1, demonstrate that MotionCache achieves significant speedups, yet delivers superior performance in perceptual quality and temporal coherence. We believe this fine-grained, motion-centric paradigm offers a promising direction for efficient video synthesis, paving the way for real-time deployment for autoregressive video generation

models.

## Acknowledgements

This work is supported by the National Key Research and Development Program of China (No. 2025YFE0113500) , the Fundamental Research Funds for the Central Universities and the National Natural Science Foundation of China ( No. 62576299).

## Impact Statement

This paper presents work whose goal is to advance the field of efficient video generation. There are many potential societal consequences of our work, none which we feel must be specifically highlighted here.

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

## A. Detailed Proof of Proposition 4.1

**Restatement of Proposition 4.1.** *The approximation error at timestep $t-1$ for chunk $i$ is strictly proportional to the magnitude of the vector difference between the true residual $\mathcal{R}_{t-1}^i$ and the cached residual $\mathcal{R}_t^i$:*

$$\epsilon_{t-1}^i = \Delta t \cdot \|\mathcal{R}_{t-1}^i - \mathcal{R}_t^i\|_2. \tag{14}$$

*Proof.* Consider the standard Euler discretization step in the flow-matching framework. For the $i$-th video chunk at timestep $t-1$, the ground-truth update using the true velocity $v_\theta(\mathbf{X}_{t-1}^i, t-1, c)$ is given by:

$$\begin{aligned}
\mathbf{X}_{t-2}^i &= \mathbf{X}_{t-1}^i + v_\theta(\mathbf{X}_{t-1}^i, t-1, c)\Delta t \\
&= \mathbf{X}_{t-1}^i + (\mathbf{X}_{t-1}^i + \mathcal{R}_{t-1}^i)\Delta t,
\end{aligned} \tag{15}$$

where $\mathcal{R}_{t-1}^i$ is the true residual derived from the model's full computation.

When the feature caching mechanism is activated, the system bypasses the computation at $t-1$ and instead reuses the residual $\mathcal{R}_t^i$ stored from the preceding timestep $t$. Consequently, the approximated update is formulated as:

$$\tilde{\mathbf{X}}_{t-2}^i = \mathbf{X}_{t-1}^i + (\mathbf{X}_{t-1}^i + \mathcal{R}_t^i)\Delta t. \tag{16}$$

The local approximation error $\epsilon_{t-1}^i$ is defined as the Euclidean distance between the ground-truth output latent $\mathbf{X}_{t-2}^i$ and the approximated latent $\tilde{\mathbf{X}}_{t-2}^i$. Substituting Eq. 15 and Eq. 16 into the error definition, we perform the subtraction:

$$\begin{aligned}
\epsilon_{t-1}^i &= \|\mathbf{X}_{t-2}^i - \tilde{\mathbf{X}}_{t-2}^i\|_2 \\
&= \left\|\left[\mathbf{X}_{t-1}^i + (\mathbf{X}_{t-1}^i + \mathcal{R}_{t-1}^i)\Delta t\right] - \left[\mathbf{X}_{t-1}^i + (\mathbf{X}_{t-1}^i + \mathcal{R}_t^i)\Delta t\right]\right\|_2 \\
&= \left\|(\mathbf{X}_{t-1}^i - \mathbf{X}_{t-1}^i) + (\mathbf{X}_{t-1}^i \Delta t - \mathbf{X}_{t-1}^i \Delta t) + (\mathcal{R}_{t-1}^i \Delta t - \mathcal{R}_t^i \Delta t)\right\|_2 \\
&= \|\Delta t \cdot (\mathcal{R}_{t-1}^i - \mathcal{R}_t^i)\|_2 \\
&= \Delta t \cdot \|\mathcal{R}_{t-1}^i - \mathcal{R}_t^i\|_2.
\end{aligned} \tag{17}$$

This derivation confirms that the error introduced by caching is linearly dependent on the step size $\Delta t$ and strictly determined by the instability of the residual vector $\mathcal{R}$ between adjacent timesteps. $\qquad\square$

## B. Detailed Proof of Lemma 4.2

**Restatement of Lemma 4.2.** *Let $\mathcal{R}(X, t)$ be the continuous residual function derived from the velocity field. Assuming the temporal gradient of the residual field $\nabla_t \mathcal{R}$ satisfies the Lipschitz condition with respect to to the input latent $X$, the residual difference across timesteps is bounded by the intra-chunk frame difference:*

$$\|\mathcal{R}_{t-1}(\mathbf{X}_{t-1}^{(i,f)}) - \mathcal{R}_t(\mathbf{X}_t^{(i,f)})\|_2 \lesssim C \cdot \|\mathbf{X}_t^{(i,f)} - \mathbf{X}_t^{(i,f-1)}\|_2, \tag{18}$$

*where $\mathbf{X}_t^{(i,f)}$ and $\mathbf{X}_t^{(i,f-1)}$ denote the latents of the $f$-th and $(f-1)$-th frames in the $i$-th chunk at timestep $t$, and $C$ is a constant.*

*Proof.* First, we analyze the term on the left-hand side, which represents the variation of the residual across discrete timesteps. By applying the first-order Taylor expansion with respect to $t$, the residual at timestep $t-1$ can be approximated as:

$$\mathcal{R}_{t-1}(\mathbf{X}_{t-1}^{(i,f)}) = \mathcal{R}_t(\mathbf{X}_t^{(i,f)}) + \frac{\partial \mathcal{R}(\mathbf{X}_t^{(i,f)}, t)}{\partial t}\Delta t + \mathcal{O}(\Delta t^2). \tag{19}$$

Ignoring higher-order terms, the magnitude of the residual difference is dominated by the partial derivative of the residual with respect to time:

$$\|\mathcal{R}_{t-1}(\mathbf{X}_{t-1}^{(i,f)}) - \mathcal{R}_t(\mathbf{X}_t^{(i,f)})\|_2 \approx \Delta t \cdot \left\|\frac{\partial \mathcal{R}(\mathbf{X}_t^{(i,f)}, t)}{\partial t}\right\|_2. \tag{20}$$

Physically, the term $\frac{\partial \mathcal{R}}{\partial t}$ corresponds to the curvature of the generative ODE trajectory. In Flow Matching models, optimal transport trajectories for static data (i.e., $\mathbf{X}_{data}^{(i,f)} = \mathbf{X}_{data}^{(i,f-1)}$) are theoretically straight lines, implying zero curvature ($\frac{\partial \mathcal{R}}{\partial t} \to 0$). Conversely, complex dynamics induce curved trajectories.

We formalize this observation by assuming that the curvature function $g(\mathbf{X}) = \frac{\partial \mathcal{R}}{\partial t}$ is Lipschitz continuous with respect to the underlying signal motion. Specifically, comparing the current frame $f$ with its adjacent frame $f-1$ (serving as a reference for local staticity):

$$\left\| g(\mathbf{X}_t^{(i,f)}) - g(\mathbf{X}_t^{(i,f-1)}) \right\|_2 \le L \cdot \|\mathbf{X}_t^{(i,f)} - \mathbf{X}_t^{(i,f-1)}\|_2, \tag{21}$$

where $L$ is the Lipschitz constant. Since the $(f-1)$-th frame serves as the immediate temporal context, for static regions where $\mathbf{X}_t^{(i,f)} \approx \mathbf{X}_t^{(i,f-1)}$, the trajectory linearity implies $g(\mathbf{X}_t^{(i,f-1)}) \approx 0$. Substituting this into Eq. 21:

$$\left\| \frac{\partial \mathcal{R}(\mathbf{X}_t^{(i,f)}, t)}{\partial t} \right\|_2 \le L \cdot \|\mathbf{X}_t^{(i,f)} - \mathbf{X}_t^{(i,f-1)}\|_2. \tag{22}$$

Finally, combining Eq. 20 and Eq. 22, we obtain the bound:

$$\|\mathcal{R}_{t-1}(\mathbf{X}_{t-1}^{(i,f)}) - \mathcal{R}_t(\mathbf{X}_t^{(i,f)})\|_2 \le (\Delta t \cdot L) \cdot \|\mathbf{X}_t^{(i,f)} - \mathbf{X}_t^{(i,f-1)}\|_2. \tag{23}$$

This concludes the proof. The caching error (residual instability) is strictly upper-bounded by the intra-chunk frame difference. $\square$

## C. Experimental Details

### C.1. Video Configuration and Model Implementation

**Video Configuration.** Regarding the specific hyperparameter settings for MotionCache, for SkyReels-V2, we set $\alpha = 0.5$ and $K = 6$. For MAGI-1, we utilize $\alpha = 0.5$ and $K = 9$. Additionally, following FlowCache (Ma et al., 2026), we designate the first $m$ timesteps as a global warm-up phase where cache reuse is disabled to ensure trajectory stability. We set $m = 5$ for MAGI-1 and $m = 4$ for SkyReels-V2.

**Architectural Differences.** While sharing an autoregressive foundation, the two models diverge fundamentally in their execution granularity. MAGI-1 operates at the inter-chunk level, utilizing a sliding window to manage the concurrent denoising of sequentially dependent chunks. Conversely, SkyReels-V2 employs a hierarchical intra-chunk strategy, subdividing chunks into granular blocks. It enforces a staggered inference schedule where earlier blocks precede subsequent ones in the denoising chain, resulting in asynchronous noise levels across the sequence at any given timestep.

### C.2. Evaluation Metrics Selection

We evaluated performance using representative VBench metrics selected based on established practices in video generation compression research (Zhao et al., 2024a; Feng et al., 2025; Shao et al., 2025). To facilitate an intuitive comparison, we compute the average scores of the selected VBench metrics using the official normalization and weighting methodology provided by the VBench benchmark.

## D. Detailed Ablation Study Results

In this section, we present the comprehensive quantitative results for the hyperparameter ablation studies discussed in the main text. Specifically, we detail the performance variations across the full sweep range of the soft-mapping floor parameter $\alpha$ and the Phase 1 duration parameter $K$.

### D.1. Full Evaluation of Soft-mapping Floor $\alpha$

Table 4 lists the complete metrics for $\alpha$ ranging from 0.0 to 1.0 with an interval of 0.1. As observed in the table, the performance metrics stabilize significantly when $\alpha$ exceeds 0.5, showing minimal variance in quality beyond this point. Conversely, lower $\alpha$ values assign insufficient importance weights to static background tokens, preventing them from reaching the update threshold. This lack of necessary updates leads to the degradation of fine-grained background details.

*Table 4.* Detailed ablation study on the soft-mapping floor parameter $\alpha$. The sweep ranges from 0.0 to 1.0 with a step of 0.1.

| $\alpha$ | PSNR ↑ | SSIM ↑ | LPIPS ↓ |
|---|---|---|---|
| 0.0 | 20.22 | 0.7944 | 0.5853 |
| 0.1 | 20.98 | 0.8254 | 0.5040 |
| 0.2 | 21.82 | 0.8591 | 0.3663 |
| 0.3 | 22.59 | 0.8880 | 0.2189 |
| 0.4 | 22.90 | 0.8950 | 0.1744 |
| 0.5 | 23.21 | 0.9037 | 0.1165 |
| 0.6 | **23.46** | 0.9093 | 0.0875 |
| 0.7 | 23.41 | 0.9072 | 0.0872 |
| 0.8 | 23.42 | 0.9095 | **0.0862** |
| 0.9 | 23.43 | 0.9082 | 0.0877 |
| 1.0 | 23.44 | **0.9105** | 0.0866 |

*Table 5.* Detailed ablation study on the duration of Phase 1 ($K$). The sweep ranges from 0 to 17 with an interval of 1.

| $K$ | PSNR ↑ | SSIM ↑ | LPIPS ↓ |
|---|---|---|---|
| 0 | 20.79 | 0.8636 | 0.1963 |
| 1 | 21.74 | 0.8902 | 0.1452 |
| 2 | 22.92 | 0.8960 | 0.1214 |
| 3 | 22.92 | 0.8993 | 0.1027 |
| 4 | 22.82 | 0.8970 | 0.1061 |
| 5 | 23.28 | 0.9054 | 0.0918 |
| 6 | **23.46** | 0.9093 | 0.0875 |
| 7 | 23.37 | 0.9096 | 0.0870 |
| 8 | 23.33 | 0.9086 | 0.0863 |
| 9 | 23.39 | 0.9090 | 0.0883 |
| 10 | 23.41 | 0.9089 | 0.0888 |
| 11 | 23.41 | 0.9094 | 0.0866 |
| 12 | 23.41 | 0.9092 | **0.0859** |
| 13 | 23.43 | **0.9106** | 0.0872 |
| 14 | 23.43 | 0.9102 | 0.0864 |
| 15 | 23.43 | 0.9103 | 0.0865 |
| 16 | 23.42 | 0.9096 | 0.0868 |
| 17 | 23.43 | 0.9103 | 0.0867 |

## D.2. Full Evaluation of Phase 1 Duration $K$

Table 5 presents the performance metrics for the Phase 1 duration $K$ ranging from 0 to 17. Notably, the setting of $K = 17$ corresponds to the FlowCache baseline, where the coarse-grained full update is applied throughout the entire generation process. As indicated by the data, the evaluation scores exhibit significant stability once $K$ exceeds 5. This trend suggests that the global semantic structure is sufficiently established by this stage, ensuring that the spatial masks can accurately align with and capture the dynamic tokens within the video.

## E. Supplementary Theoretical Analysis for Lemma 4.2

This section provides additional theoretical justification for the Lipschitz assumption and the validity of omitting higher-order terms in the Taylor expansion used in Lemma 4.2.

### E.1. Justification of the Lipschitz Assumption

The Lipschitz continuity of the residual temporal gradient $\nabla_t \mathcal{R}(\mathbf{X}, t)$ with respect to the input latent $\mathbf{X}$ is a theoretically grounded property of modern Diffusion Transformer (DiT) backbones.

Formally, let $g(\mathbf{X}) = \nabla_t \mathcal{R}(\mathbf{X}, t)$ denote the curvature function of the generative trajectory. For a DiT backbone with sinusoidal time embedding, the velocity field can be expressed as:

$$v_\theta(\mathbf{X}, t) = \mathcal{F}_\theta(\mathbf{X}, e(t)), \tag{24}$$

where $e(t) = [\sin(\omega_k t), \cos(\omega_k t)]_{k=1}^K$ is the sinusoidal time embedding with bounded frequencies $\omega_k$. The residual function is then:

$$\mathcal{R}(\mathbf{X}, t) = v_\theta(\mathbf{X}, t) - \mathbf{X}. \tag{25}$$

By the chain rule, the temporal gradient of the residual is:

$$\frac{\partial \mathcal{R}}{\partial t} = \frac{\partial \mathcal{F}_\theta}{\partial e} \cdot \frac{de(t)}{dt}. \tag{26}$$

The Lipschitz constant of $g(\mathbf{X})$ is bounded by the product of two terms: 1. The spectral norm of the Jacobian $\left\| \frac{\partial \mathcal{F}_\theta}{\partial e} \right\|_2$, which is guaranteed to be finite by the weight normalization and spectral regularization inherent in modern transformer training. 2. The magnitude of the time embedding derivative $\left\| \frac{de(t)}{dt} \right\|_2$, which is bounded by the maximum frequency $\max(\omega_k)$ in the sinusoidal embedding.

Since both terms are bounded by design, the function $g(\mathbf{X})$ satisfies the Lipschitz condition with constant $L = \left\| \frac{\partial \mathcal{F}_\theta}{\partial e} \right\|_2 \cdot \max(\omega_k)$.

### E.2. Validity of Omitting Higher-Order Terms

The omission of the $\mathcal{O}(\Delta t^2)$ term in the Taylor expansion is empirically justified by the significant magnitude gap between the first-order term and higher-order terms in practical diffusion schedules.

For a typical 50-step denoising schedule used in our experiments, the time step size is $\Delta t = 1/50 = 0.02$, resulting in $\Delta t^2 = 0.0004$. In contrast, as shown in Figure 2 (a), the magnitude of the residual difference $\|\mathcal{R}_{t-1} - \mathcal{R}_t\|_2$ is at least 1.175 across all timesteps. This means the higher-order term is at least three orders of magnitude smaller than the first-order term, making it negligible in practice.

## F. Peak Memory and FLOPs Analysis for Long Video Generation

To validate the memory stability and computational efficiency of MotionCache in long video scenarios, we supplement the peak memory and FLOPs breakdown analysis for both the original 7s setting and the 10s setting.

### F.1. Key Observations

From the detailed results presented below, we draw the following conclusions: 1.The main FLOPs cost consistently comes from attention operations (self + cross), which scale quadratically with sequence length and dominate the overall computation. 2.Compared with other methods, the peak memory increase introduced by MotionCache remains stable as video length increases from 7s to 10s, and is acceptable in practice. 3.MotionCache consistently maintains lower overall computation than all baselines across both video lengths, demonstrating its superior scalability.

Taken together, these results show that MotionCache remains applicable and effective in long-video generation scenarios.

### F.2. Detailed Results on SkyReels-V2

Table 6 presents the peak memory and FLOPs breakdown for SkyReels-V2 on 7s and 10s videos.

Notably, MotionCache maintains the same peak memory usage (42GB) as FlowCache for both 7s and 10s videos, while achieving lower total FLOPs.

*Table 6.* Peak memory and FLOPs breakdown for SkyReels-V2 (7s vs 10s)

| Length | Method | Peak Memory (GB) | Total TFLOPs | Attn (Self+Cross) | AttnGEMM | FFNGEMM |
|---|---|---|---|---|---|---|
| 7s | vanilla | 34 | 113 | 79 | 12 | 22 |
| | teacache | 36 | 58 | 40 | 6 | 12 |
| | flowcache | 42 | 31 | 22 | 3 | 6 |
| | motioncache | 42 | 30 | 21 | 3 | 6 |
| 10s | vanilla | 34 | 168 | 117 | 17 | 34 |
| | teacache | 36 | 72 | 50 | 7 | 15 |
| | flowcache | 42 | 47 | 33 | 5 | 9 |
| | motioncache | 42 | 40 | 28 | 4 | 8 |

## F.3. Detailed Results on MAGI-1

Table 7 presents the peak memory and FLOPs breakdown for MAGI-1 on 7s and 10s videos.

*Table 7.* Peak memory and FLOPs breakdown for MAGI-1 (7s vs 10s)

| Length | Method | Peak Memory (GB) | Total TFLOPs | Attn (Self+Cross) | AttnGEMM | FFNGEMM |
|---|---|---|---|---|---|---|
| 7s | vanilla | 26.41 | 139 | 84 | 20 | 35 |
| | teacache | 26.66 | 129 | 78 | 18 | 33 |
| | flowcache | 22.45 | 104 | 63 | 15 | 26 |
| | motioncache | 22.45 | 100 | 60 | 14 | 26 |
| 10s | vanilla | 31.39 | 207 | 128 | 29 | 50 |
| | teacache | 31.64 | 195 | 121 | 27 | 47 |
| | flowcache | 27.43 | 145 | 90 | 20 | 35 |
| | motioncache | 27.43 | 142 | 89 | 19 | 34 |

# G. Analysis of RGB-domain Optical Flow Synergy

To address the question of whether optical flow in the RGB domain can work synergistically with our current acceleration mechanisms, we provide a detailed analysis of the computational trade-offs and architectural compatibility.

## G.1. Architectural Incompatibility

Our MotionCache framework operates entirely in the VAE-compressed latent space, which is critical for its high efficiency. In contrast, RGB-domain optical flow must be computed on raw pixel data. This would require decoding the latent tokens back to RGB at every denoising step, which is prohibitively expensive in practice.

For example, on SkyReels-V2, a single VAE decode operation for one video chunk takes 9.7 seconds on an NVIDIA A800 80GB GPU. Since our method performs caching across dozens of timesteps, the cumulative overhead of repeated VAE decoding would completely negate any acceleration benefits from caching.

## G.2. Computational Overhead Comparison

We further compared the inference time of our proposed frame difference motion cue with two standard optical flow methods: sparse optical flow (KLT) and dense optical flow (Farneback).

As shown in Table 8, both sparse and dense optical flow are 429× and 925× slower than our simple frame difference signal, respectively. This makes them impractical for efficient acceleration, as their computational overhead would far exceed the savings from caching.

*Table 8.* Computational time comparison of different motion cues

| Motion Cue | Time (ms) |
| --- | --- |
| Frame Difference (Ours) | 3.46 |
| Sparse Optical Flow (KLT) | 1483 |
| Dense Optical Flow (Farneback) | 3201 |

## H. More Qualitative Results

In this section, we provide extensive qualitative visualizations to further validate the effectiveness of MotionCache. We first visualize the temporal evolution of the motion-aware importance maps to justify our coarse-to-fine schedule. Subsequently, we present visual comparisons of the actual generated videos across different methods on SkyReels-V2 and MAGI-1.

### H.1. Evolution of Motion Importance Maps

To justify the necessity of the proposed Dual-Stage Coarse-to-Fine Inference Schedule (Sec. 5.3), we visualize the importance weight maps $\mathcal{W}$ throughout the denoising process in Figure 6.

As observed in the early timesteps, the importance weights exhibit a diffuse and unstructured distribution. At this stage, the global semantic layout is not yet stabilized, and the model cannot effectively distinguish between foreground motion and static background. Consequently, a rigid chunk-wise update (Phase 1) is crucial here to ensure structural integrity. As the denoising progresses, the importance maps become increasingly sparse and structured, precisely highlighting the dynamic contours of the moving subject. This clear separation validates the transition to Phase 2, where our token-wise caching strategy efficiently allocates computation to these high-motion regions.

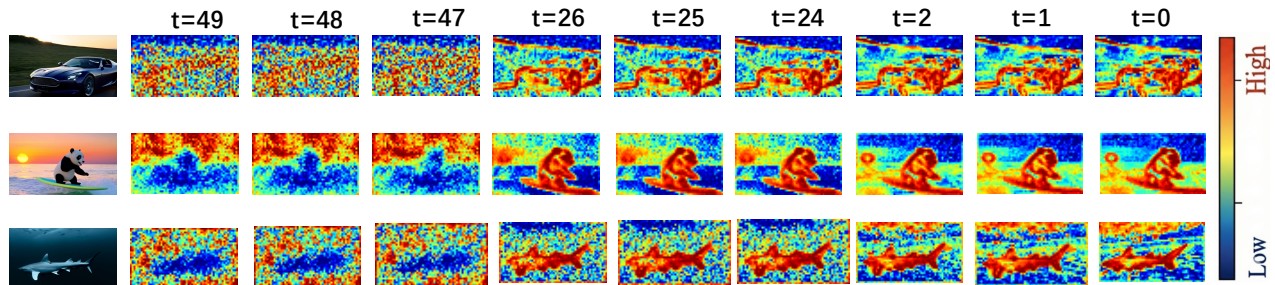

*Figure 6.* Visualization of the importance weight maps $\mathcal{W}$ throughout the denoising process. The label $t$ indicates the denoising timestep. The leftmost column displays the final ground-truth video frames. In the early inference stages, the weight distribution remains diffuse and unstructured with ambiguous contours, indicating that the global semantic structure is not yet clearly established. As generation proceeds, the maps sharpen to accurately capture motion dynamics.

### H.2. Qualitative Comparison on SkyReels-V2

Figure 7 presents a visual comparison on SkyReels-V2. While TeaCache provides a $2.2\times$ speedup, it suffers from noticeable high-frequency noise, particularly evident in the astronaut and beer scenarios. FlowCache achieves a significant speedup of $6.26\times$ but is prone to semantic misalignment and content drift; for instance, the cyclist's sleeve texture is missing, and the person tasting beer exhibits anatomical hallucinations (e.g., six fingers). In contrast, our MotionCache achieves a comparable high speedup of $6.28\times$ while preserving superior visual fidelity. It effectively maintains structural consistency with the Vanilla baseline and achieves the highest PSNR.

### H.3. Qualitative Comparison on MAGI-1

Figure 8 illustrates the visual results on MAGI-1. Similar to SkyReels-V2, MotionCache demonstrates a superior ability to maintain high fidelity to fine-grained semantic details that are often lost during acceleration. A striking example is seen in the "elephant" sequence: while other methods result in the complete disappearance of the elephant's tusks, MotionCache successfully preserves them by accurately identifying and updating these critical regions. Furthermore, our method maintains

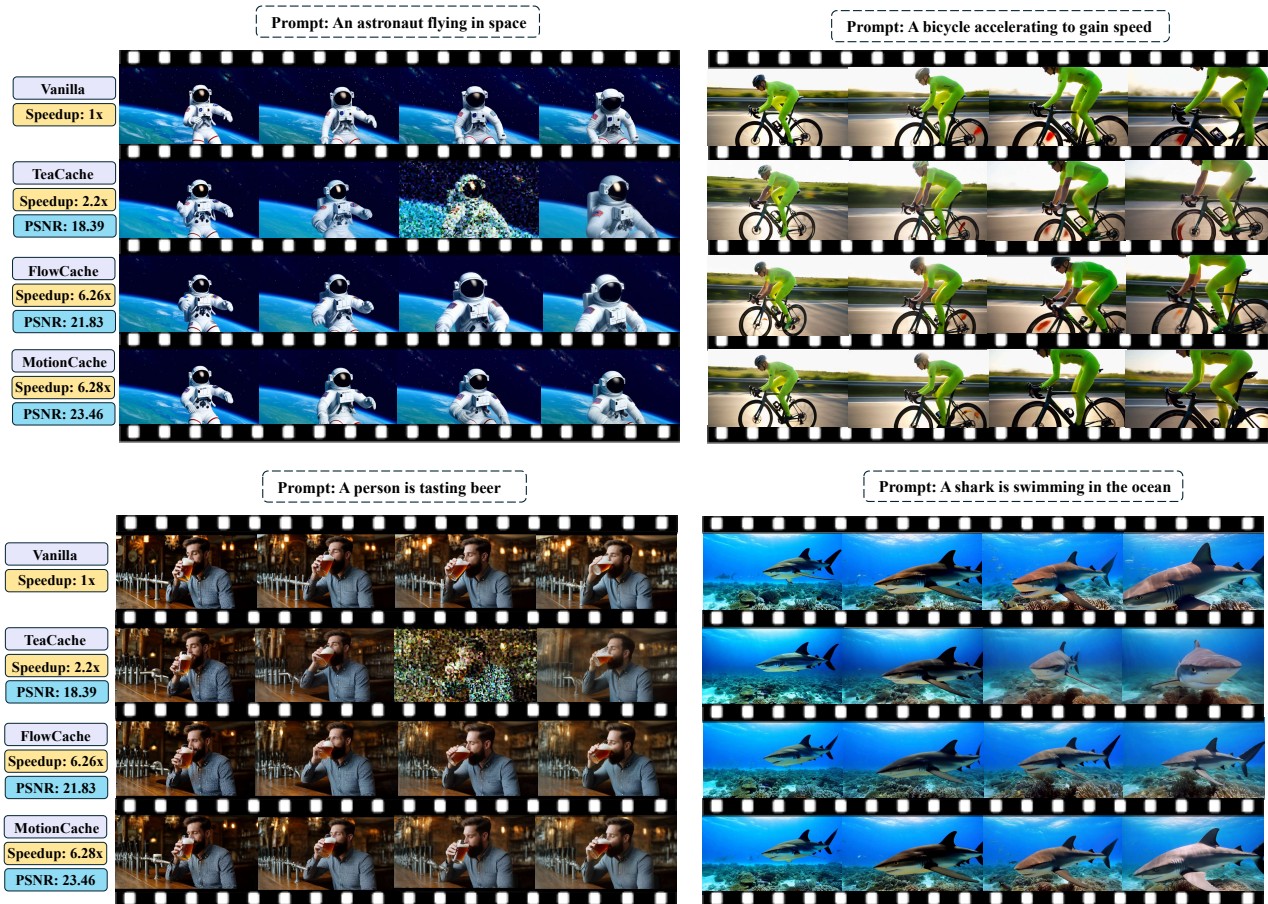

*Figure 7.* Qualitative results of text-to-video generation on SkyReels-V2. We present TeaCache, FlowCache, MotionCache, and the Vanilla model. The frames are randomly sampled from the generated video.

a consistent and natural horse color throughout the video, avoiding the color bleeding and flickering artifacts prevalent in TeaCache and FlowCache. These qualitative improvements underscore that MotionCache's token-wise precision is essential for preserving the structural and aesthetic integrity of complex subjects.

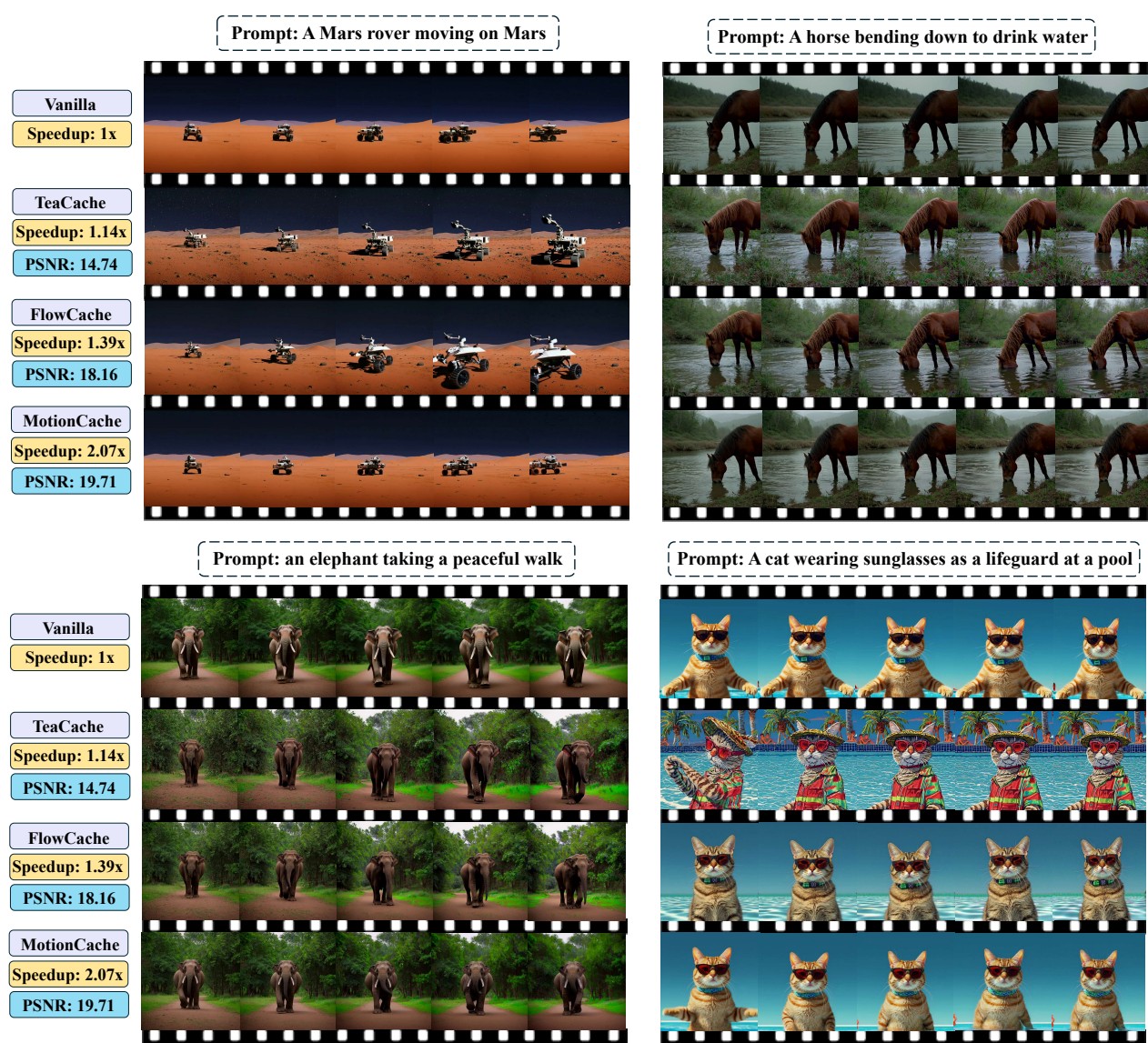

*Figure 8.* Qualitative results of text-to-video generation on MAGI-1. We present TeaCache, FlowCache, MotionCache, and the Vanilla model. The frames are randomly sampled from the generated video.

