# OpenReview forum: "Motion-Aware Caching for Efficient Autoregressive Video Generation"
_ICML.cc/2026/Conference — ICML 2026 regular_

### Official Review · Reviewer_w7Qa · 2026-02-26

**Soundness:** 3
**Presentation:** 3
**Significance:** 3
**Originality:** 3
**Overall Recommendation:** 4
**Confidence:** 4

**Summary:**

The paper proposes motion-cache algorithm, a motion-aware cache framework that exploits inter-frame differences as a lightweight proxy for pixel-level motion characteristics. The proposed method employs a coarse-to-fine strategy to dynamically adjusts update frequencies per token, The experiments show that the proposed motion-cache achieves significant speedups.

**Compliance With Llm Reviewing Policy:**

Affirmed.

**Final Justification:**

I have read the rebuttal and my concerns have been addressed.

**Key Questions For Authors:**

How to efficiently implement partial calculation? In this paper, equation 13, the author use a indicator function to construct a mask for partial calculation. However, in real system, a random indexed mask is not hardware friendly, the index selection and  not continuous memory usage could slow down the calculation process. How did the author consider this issue?

**Limitations:**

The author should adds more baselines for comparison, only Tea-cache and Flow-cache is not enough.  Some related methods such as  ToCA, TaylorSeer can also be adapted.

**Strengths And Weaknesses:**

Strengths:
    The motivation of this paper is strong. Before introducing the proposed motion cache framework, the author presents solid analysis and cues for the reason of latent difference can reflect the motion of pixels. Then use a coarse-to-fine strategy to decide which chunk for full computation, which chuck to skip calculation, which chuck to perform partial computation, the whole framework is reasonable.
 Weaknesses:
    The author should adds more baselines for comparison, only Tea-cache and Flow-cache is not enough.  Some related methods such as  ToCA, TaylorSeer can also be adapted.

---

> ### Author Rebuttal · Authors · 2026-03-31
>
> **W1: The author should adds more baselines for comparison, only Tea-cache and Flow-cache is not enough. Some related methods such as ToCA, TaylorSeer can also be adapted.**
>
> We appreciate the reviewer's suggestion. Following this comment, we further adapted ToCA and TaylorSeer as additional baselines and evaluate them on SkyReels-V2.
>
> The experiments follow the same video configuration as in our paper, i.e., 540p resolution, 2 chunks, 97 frames per chunk, and 24 FPS. Except for the acceleration method, all other video generation settings are kept the same.
>
> | Method | Peak Memory | Speedup | PSNR (↑) | SSIM (↑) | LPIPS (↓) |
> |---|---:|---:|---:|---:|---:|
> | Vanilla | 34GB | - | - | - | - |
> | TeaCache | 36GB | 2.24× | 18.39 | 0.6121 | 0.3063 |
> | ToCA | 48GB | 2.18× | 20.81 | 0.8186 | 0.4423 |
> | TaylorSeer | 72GB | 2.24× | 20.24 | 0.8433 | 0.2136 |
> | FlowCache | 42GB | 6.26× | 21.83 | 0.8733 | 0.1417 |
> | MotionCache | 42GB | **6.28×** | **23.46** | **0.9093** | **0.0875** |
>
> Among all compared methods, MotionCache still achieves the best overall trade-off between efficiency and quality.
>
> For reproducibility, we use interval = 3 and sparsity = 60% for ToCA. For TaylorSeer, we use O = 1 and interval = 3; using O = 2 leads to OOM under our evaluation setting.
>
> **Q1: How to efficiently implement partial calculation? In this paper, equation 13, the author use a indicator function to construct a mask for partial calculation. However, in real system, a random indexed mask is not hardware friendly, the index selection and not continuous memory usage could slow down the calculation process. How did the author consider this issue?**
>
> We agree that random indexed masking is generally less hardware-friendly. In our implementation, the index-related overhead mainly comes from three operations: identifying the indices of tokens to be updated and select those tokens, updating the KV cache according to these indices, and writing the updated token features back to the output according to the indices. We profiled these index-related operations and found that their total overhead is only **0.415 s** over the whole inference process, compared with **245 s** end-to-end inference time. This indicates that the practical overhead of index manipulation is negligible in practice.
>
> | Component | Time (s) |
> |---|---:|
> | Indexing  mask overhead | 0.415 |
> | Total end-to-end time | 245 |

---

> > ### Author Rebuttal · Reviewer_w7Qa · 2026-04-02
> >
> > Fully resolved

---

> > > ### Author Response · Authors · 2026-04-07
> > >
> > > Thank you again for your thoughtful review and follow-up. We are glad to know that our rebuttal has fully addressed your concerns.
> > >
> > > If you find it appropriate, we would sincerely appreciate it if you could kindly reconsider the current score in light of the additional experiments and clarifications provided in the rebuttal. In any case, we truly appreciate your time and careful evaluation of our work.

---

### Official Review · Reviewer_1wX8 · 2026-03-12

**Soundness:** 3
**Presentation:** 3
**Significance:** 3
**Originality:** 2
**Overall Recommendation:** 3
**Confidence:** 4

**Summary:**

The paper proposes MotionCache, which first analyzes cache error through residual instability, then uses intra-chunk frame differences as a lightweight proxy for motion, and finally applies a coarse-to-fine inference schedule.

**Compliance With Llm Reviewing Policy:**

Affirmed.

**Final Justification:**

I still feel that the novelty is moderate, so I will keep my score unchanged.

**Key Questions For Authors:**

1. Can you distinguish how much of the performance improvement stems from motion sensing itself versus switching from block caching to tag caching?
2. Can you strengthen the evidence behind Lemma 4.2 and the motion proxy?
3. Is the comparison with FlowCache fair and comprehensive?
4. Can you provide a more thorough evaluation , ideally incorporating the full VBench testing dimensions, along with additional evidence of model generalization to other autoregressive models or longer videos?

**Limitations:**

Please refer to the weakness part.

**Strengths And Weaknesses:**

Strength:
1.The paper tackles a important problem in autoregressive video generation, and the proposed method is technically sound.

2. The paper is generally clear.

Weakness:
1.The theoretical section is only partially convincing. Lemma 4.2 is interesting, but it relies on fairly strong assumptions and ignores higher-order terms. Consequently, the assertion that “frame error is an upper bound on cache error” appears more like an intuitive approximation than a rigorous guarantee.

2. The improvement over FlowCache is not substantial.

3. The novelty is moderate.  Some core elements already exist in related work.

---

> ### Author Rebuttal · Authors · 2026-03-31
>
> **W1 & Q2: On the Rigor of Lemma 4.2 and Higher-order Terms.**
> We thank the reviewer for the constructive comment, which prompted us to provide justification of the assumption and analysis of higher-order terms.
> **1.The Lipschitz Assumption.**
> The $L$-Lipschitz continuity of $F(\mathbf{X}) = \nabla\_t \mathcal{R}(\mathbf{X}, t)$ follows from bounding its spatial Jacobian. By the Mean Value Theorem, for any $\mathbf{X}\_1, \mathbf{X}\_2$, there exists intermediate point $\mathbf{C}$ such that $F(\mathbf{X}\_1) - F(\mathbf{X}\_2) = J\_F(\mathbf{C})(\mathbf{X}\_1 - \mathbf{X}\_2)$, which implies $\|F(\mathbf{X}\_1) - F(\mathbf{X}\_2)\|\_2 \le \|J\_F(\mathbf{C})\|\_2 \cdot \|\mathbf{X}\_1 - \mathbf{X}\_2\|\_2$. Thus, the Lipschitz condition holds if $\|J\_F(\mathbf{X})\|\_2 = \|\nabla\_{\mathbf{X}}\nabla\_t v\_\theta(\mathbf{X}, t)\|\_2 \le L$.
>
> In DiT backbones where $v\_\theta = \mathcal{F}\_\theta(\mathbf{X}, e(t))$, the chain rule yields $\nabla\_{\mathbf{X}}\nabla\_t v\_\theta = \frac{\partial^2 \mathcal{F}\_\theta}{\partial \mathbf{X} \partial e} \cdot \frac{de(t)}{dt}$. Both factors are bounded by design: $\|\frac{de}{dt}\|\_2 \le C\_t$ is limited by finite frequencies $\omega\_k$ in the sinusoidal embedding, while $\|\frac{\partial^2 \mathcal{F}\_\theta}{\partial \mathbf{X} \partial e}\|\_2 \le C\_\theta$ is guaranteed by finite weight spectral norms.
>
> **2.Omitting Higher-order Terms.** The omission of $\mathcal{O}(\Delta t^2)$ is justified by the magnitude gap. From the Taylor expansion $\mathcal{R}\_{t-1}(\mathbf{X}\_{t-1}) = \mathcal{R}\_t(\mathbf{X}\_t) + \frac{\partial \mathcal{R}}{\partial t} \Delta t + \mathcal{O}(\Delta t^2)$, a typical 50-step schedule gives $\Delta t = 0.02$ and thus $\Delta t^2 = 0.0004$. Meanwhile, Figure 2(a) shows that the residual difference $\|\mathcal{R}\_{t-1} - \mathcal{R}\_t\|\_2$ is at least $1.175$. This makes the higher-order term negligible in practice.
>
> **W2: Comparison with FlowCache.**
> We believe the improvement over FlowCache is meaningful for two reasons.
>
> First, under high acceleration ratios, large VBench gaps are difficult to obtain. In prior works, the reported improvements are often relatively small, e.g., around 0.05% for SpeCa [1], 0.18% for ClusCa [2], and even -0.01% for MeanCache [3].
>
> Second, our advantage is more clearly reflected in video consistency. On SkyReels-V2, MotionCache achieves higher PSNR (23.46 vs. 21.83) and SSIM (0.9093 vs. 0.8733), and lower LPIPS (0.0875 vs. 0.1417). This is due to its motion-aware design, which allocates computation to dynamic regions and better preserves the main subject.
>
> **W3: Novelty of the proposed method.**
> We appreciate the reviewer's concern on novelty.
>
> First, while the general intuition of using motion dynamics is not entirely new, prior works do not explain why motion reflects caching error or how it can be exploited for token-level caching.
>
> Second, we analyze heterogeneous temporal redundancy and intra-chunk frame discrepancy, connect caching error to residual stability, and show that intra-chunk frame difference is an effective motion-aware importance signal. Based on this, we design MotionCache with a coarse-to-fine schedule and an importance-weighted token accumulation policy.
>
> Overall, our novelty lies not only in individual components, but also in a complete motion-aware caching framework.
>
> **Q1: Contribution of motion sensing and tag caching.**
> To isolate motion sensing, we keep the same token-wise caching framework and only vary the importance metric; detailed results are provided at https://anonymous.4open.science/r/MotionCache/table%201.pdf.
>
> Using frame difference consistently outperforms mean and std, showing that the gain comes from both fine-grained token caching and motion sensing.
>
> **Q3: Fair Comparison with FlowCache**
>
> Our comparison with FlowCache is fair and comprehensive: all methods are conducted using PyTorch on NVIDIA A800 80GB GPUs, and with exactly the same generation settings. We also follow common practice in video generation acceleration researchs[1,2,3] by reporting VBench, PSNR, SSIM, and LPIPS for quality, and FLOPs with end-to-end latency for efficiency.
>
> **Q4: Full VBench Evaluation, Generalization to Other Autoregressive Models, and Results on Longer Videos**
>
> We have provided the full VBench dimension scores for all configurations at https://anonymous.4open.science/r/MotionCache/table%202.pdf.
>
> Extending to other autoregressive models requires re-adapting each method, which we leave to future work due to limited time and resources. For longer videos, please refer to our response to Reviewer cJeY, W1.
>
> **References**
> [1]Speca: Accelerating diffusion transformers with speculative feature caching. ACM MM 2025.
> [2]Compute only 16 tokens in one timestep: Accelerating diffusion transformers with cluster-driven feature caching.ACM MM 2025.
> [3]MeanCache: From Instantaneous to Average Velocity for Accelerating Flow Matching Inference. ICLR 2026.

---

> > ### Author Rebuttal · Reviewer_1wX8 · 2026-04-05
> >
> > My concerns are paritally solved.

---

> > > ### Author Response · Authors · 2026-04-06
> > >
> > > Thank you for your follow-up. We are glad to hear that our rebuttal partially addressed your concerns.
> > >
> > > To help us better understand your remaining reservations, could you please clarify which part you still have concerns about?
> > >
> > > Thank you again for your time and feedback.

---

### Official Review · Reviewer_cJeY · 2026-03-13

**Soundness:** 3
**Presentation:** 3
**Significance:** 2
**Originality:** 2
**Overall Recommendation:** 4
**Confidence:** 3

**Summary:**

This paper identifies the limitation of traditional coarse-grained caching strategies in capturing pixel-level motion dynamics and avoiding error accumulation. It establishes a mathematical link between caching error, residual instability and video motion via theoretical derivations, validating intra-chunk inter-frame latent feature differences as a lightweight motion proxy. Based on this, MotionCache is proposed, which enables dynamic token-level computational resource allocation through motion-aware token importance scoring, weight-guided error accumulation and a two-stage coarse-to-fine inference schedule. Experiments on SkyReels-V2 and MAGI-1 show that MotionCache achieves speedups of 6.28×/1.64× (slow) and 7.26×/2.07× (fast) with only 1%/0.01% VBench degradation, outperforming TeaCache and FlowCache on PSNR, SSIM and other metrics while maintaining high visual consistency with the vanilla model. It provides an effective solution for efficient autoregressive video generation inference.

**Compliance With Llm Reviewing Policy:**

Affirmed.

**Key Questions For Authors:**

1. Supplement peak memory and FLOPs analysis for long video generation to validate memory stability, and explain its adaptability and boundaries for long video scenarios in the discussion.
2. Can optical flow in the RGB domain work synergistically with current acceleration mechanisms?

**Limitations:**

Yes

**Strengths And Weaknesses:**

Strength
1. The first work to rigorously derive caching approximation error sources, proposing two key principles that mathematically correlate motion dynamics with caching error risk. This elevates motion-aware caching from empirical heuristic to theoretically grounded decision-making, filling the theoretical gap of existing methods.
Precise design with essential upgrades: Breaking the block-level limitation of FlowCache and other methods, it realizes fine-grained token-level motion-aware caching. The two-stage inference schedule aligns with the autoregressive generation rule (structure first, detail later), balancing global semantic consistency and acceleration efficiency for an essential optimization over block-level strategies.
2. Validated on two SOTA models with different resolutions/architectures, with slow/fast configurations for quality-acceleration trade-off analysis. Quantitative evaluation (pixel, perceptual, video-specific metrics) and qualitative visualizations fully demonstrate its superiority. Detailed ablation on core hyperparameters clarifies optimal settings and robustness, enhancing reproducibility.
3. Motion awareness is realized with the model’s native latent features, requiring no additional optical flow models/branches and incurring no extra computation/memory overhead. Robust hyperparameters enable plug-and-play integration across models without elaborate tuning, meeting real-world deployment needs.

Weakness
* Lack of comparative analysis with FlowCache (long video-focused) in long video scenarios, failing to verify its core advantages directly.

---

> ### Author Rebuttal · Authors · 2026-03-31
>
> **W1: Lack of comparative analysis with FlowCache in long-video scenarios.**
>
> We appreciate the reviewer's suggestion and have conducted additional long-video experiments to provide a more direct comparison with FlowCache. Specifically, we extend the generation length from **7s** to **10s** and evaluate both methods in this longer setting. In addition, videos longer than 5s are already considered long videos in VBench, and our evaluation follows the VBench-long protocol.
>
> The results are summarized below. Even in the 10s setting, MotionCache still shows the best overall performance. In particular, it performs well on motion_smoothness and scene, mainly due to more updates being allocated to dynamic motion regions.
>
> **SkyReels-V2 (10s)**
>
> | Method | TFLOPs | Vbench | subject_consistency | background_consistency | motion_smoothness | aesthetic_quality | imaging_quality | scene | overall_consistency |
> |---|---:|---:|---:|---:|---:|---:|---:|---:|---:|
> | vanilla | 168 | 83.13% | 95.2% | 96.25% | 98.64% | 64.89% | 65.56% | 50.68% | 26.58% |
> | Teacache | 72 | 78.29% | 87.95% | 93.02% | 97.86% | 57.89% | 56.81% | 39.53% | 26.46% |
> | Flowcache | 47 | 82.34% | 95.41% | 96.19% | 98.73% | 64.32% | 64.05% | 47.76% | 26.56% |
> | Motioncache | 40 | 82.65% | 95.37% | 96.09% | 98.74% | 64.42% | 63.91% | 51.06% | 26.58% |
>
>
> **MAGI-1 (10s)**
>
> | Method | TFLOPs | Vbench | subject_consistency | background_consistency | motion_smoothness | aesthetic_quality | imaging_quality | scene | overall_consistency |
> |---|---:|---:|---:|---:|---:|---:|---:|---:|---:|
> | vanilla | 207 | 78.22% | 97.81% | 98.03% | 99.28% | 61.81% | 64.20% | 32.86% | 26.23% |
> | Teacache | 195 | 76.83% | 97.03% | 97.78% | 98.19% | 61.22% | 64.26% | 27.67% | 25.97% |
> | Flowcache | 145 | 77.62% | 97.22% | 97.72% | 98.58% | 62.13% | 64.58% | 28.90% | 26.79% |
> | Motioncache | 142 | 78.21% | 98.57% | 98.31% | 99.54% | 61.46% | 63.87% | 30.68% | 26.18% |
>
> **Q1: Peak memory and FLOPs analysis for long video generation and its adaptability and boundaries.**
>
> We supplement the peak memory and FLOPs analysis for both the original 7s setting and the extended 10s setting. The detailed results are reported below.
>
> From these results, we draw the following conclusions.
> 1. The main FLOPs cost consistently comes from attention (self + cross), which scales quadratically with sequence length and dominates the overall computation.
> 2. Compared with other methods, the peak memory increase introduced by MotionCache remains stable in longer videos and is acceptable in practice, while still maintaining lower overall computation.
>
> Taken together, and combined with the VBench results in W1, these results show that MotionCache remains applicable and effective in long-video generation scenarios.
>
> **SkyReels-V2**
>
> | Length | Method | Peak Memory | Total TFLOPs | Attn (Self + Cross) | AttnGEMM | FFNGEMM |
> |---|---|---:|---:|---:|---:|---:|
> | 7s | vanilla | 34GB | 113 | 79 | 12 | 22 |
> | 7s | teacache | 36GB | 58 | 40 | 6 | 12 |
> | 7s | flowcache | 42GB | 31 | 22 | 3 | 6 |
> | 7s | motioncache | 42GB | 30 | 21 | 3 | 6 |
> | 10s | vanilla | 34GB | 168 | 117 | 17 | 34 |
> | 10s | teacache | 36GB | 72 | 50 | 7 | 15 |
> | 10s | flowcache | 42GB | 47 | 33 | 5 | 9 |
> | 10s | motioncache | 42GB | 40 | 28 | 4 | 8 |
>
> **MAGI-1**
>
> | Length | Method | Peak Memory | Total TFLOPs | Attn (Self + Cross) | AttnGEMM | FFNGEMM |
> |---|---|---:|---:|---:|---:|---:|
> | 7s | vanilla | 26.41GB | 139 | 84 | 20 | 35 |
> | 7s | teacache | 26.66GB | 129 | 78 | 18 | 33 |
> | 7s | flowcache | 22.45GB | 104 | 63 | 15 | 26 |
> | 7s | motioncache | 22.45GB | 100 | 60 | 14 | 26 |
> | 10s | vanilla | 31.39GB | 207 | 128 | 29 | 50 |
> | 10s | teacache | 31.64GB | 195 | 121 | 27 | 47 |
> | 10s | flowcache | 27.43GB | 145 | 90 | 20 | 35 |
> | 10s | motioncache | 27.43GB | 142 | 89 | 19 | 34 |
>
> **Q2: Can optical flow in the RGB domain work synergistically with current acceleration mechanisms?**
>
> We agree that RGB-domain optical flow could be explored as an additional motion cue. However, it is not well suited to our current design, since our method operates in the **VAE-compressed latent space**, while optical flow must be computed in the **RGB domain**. This would require decoding the latent tokens back to RGB at each step, which is prohibitively expensive in practice (the **VAE decode alone takes 9.7s** for a single chunk in SkyReels-V2).
>
> We also compare different optical-flow-based alternatives. As shown below, both sparse and dense optical flow are far more expensive than our simple **frame difference** signal, making them impractical for efficient acceleration.
>
> | Motion Cue | Time (ms) |
> |---|---:|
> | frame difference | **3.46** |
> | sparse optical flow (KLT) | 1483 |
> | dense optical flow (Farneback) | 3201 |

---

> > ### Author Rebuttal · Reviewer_cJeY · 2026-04-04
> >
> > I believe the author's additional experiments have clarified my misunderstanding.

---

> > > ### Author Response · Authors · 2026-04-07
> > >
> > > Thank you again for your time and thoughtful feedback. We are glad that our additional experiments and clarifications have addressed your concerns and helped clarify the misunderstanding.
> > >
> > > If you feel the rebuttal has satisfactorily resolved the main issues you raised, we would be very grateful if you could kindly reconsider the current rating in light of the updated evidence. Of course, we fully respect your final judgment, and we sincerely appreciate your careful evaluation of our work.

---

### Decision · Program_Chairs · 2026-04-30

**Decision:**

Accept (regular)

**Comment:**

This paper proposes MotionCache, a motion-aware token-level caching framework for accelerating autoregressive video generation. The core contribution is a theoretical grounding of caching error in terms of residual instability and motion dynamics. It is an extension from block-level cacheing methods to fine-grained token-level caching.

3 reviewers participated in the paper assessment.
Two reviewers (weak accept) found the motivation strong, the design principled, and the theoretical framing meaningful beyond empirical heuristics. They had initial concerns of the lack of comparative analysis that they acknowledged to be fully resolved after the rebuttal.
A reviewer (weak reject) kept the weak reject assessment after the rebuttal citing the moderate novelty, pointing out that the core elements (motion-aware caching, token-level sparsity) exist in prior works. The authors claim their contribution in the theoretical grounding explaining why motion reflects caching error, analysis on heterogenous temporal redundancy, and the overall framework rather than the components.

On balance, the theoretical contribution, comprehensive ablation study and plug-and-play design support weak acceptance. The authors are encouraged to strengthen the discussion of novelty relative to FlowCache and to extend generalization experiments.